# Spatial–Temporal Influence of Sand Dams on Chemical and Microbial Properties of Water from Scooping Holes in Degraded Semi-Arid Regions

Harrison Churu [1,*], Solomon Kamau [2], Wilson Ngetich [1], Keziah Magiroi [3], Bonface Alkamoi [4], Syphyline Kebeney [2], Fred Wamalwa [5] and James Mumo [1]

1   Department of Soil Science, University of Eldoret, Eldoret P.O. Box 1125-30100, Kenya
2   Department of Agriculture and Natural Resources, Moi University, Eldoret P.O. Box 3900-30100, Kenya
3   Kenya Agricultural and Livestock Research Organization (KALRO), Kitale P.O. Box 450-30200, Kenya
4   Department of Seed Crops and Horticultural Sciences, University of Eldoret,
    Eldoret P.O. Box 1125-30100, Kenya; bonnie.alka@gmail.com
5   Department of Rural Development and Agricultural Economics, University of Eldoret,
    Eldoret P.O. Box 1125-30100, Kenya
*   Correspondence: harrison.churu@uoeld.ac.ke

**Abstract:** Communities in semi-arid lands use sand dams to enhance access water during the dry seasons. However, there is limited information on the quality of water derived from these sand dams, especially in degraded lands where storm surface runoff poses contamination risk. Thus, this study aimed at assessing the spatial–temporal variations in water quality of sand dams in Chepareria, West Pokot County in Kenya. Water samples were collected from scooping holes across 18 purposefully selected sand dams. Results obtained showed significant differences in water quality based on a sand dam's age and location of the scooping holes, but the magnitude of these differences differed with specific properties. For instance, in recently constructed sand dams (<1 year), scooping holes near the sand dam wall had lower pH values (8.5) than holes scooped a distance from the sand dam wall (9.2). For total dissolved solutes and microbial properties, sand dam age had the greatest impact, over the location of the scooping holes. For example, water obtained from <1 year old sand dams had significantly higher TDS with an average value of 100.3 mg L$^{-1}$. The thermotolerant coliforms (TTC) exceeded the maximum allowable levels recommended by The World Health Organization. Thus, water obtained from these sand dams should be treated before consumption. Finally, sand dams meant for domestic water harvesting should be protected. Shallow wells with appropriate aprons for effective protection against contamination should be installed to enhance abstraction of safe water from sand dams.

**Keywords:** fecal coliforms; scooping holes; water quality

## 1. Introduction

Access to clean, safe and adequate water is a key developmental goal enshrined in the sustainable development goals (SGD 6) and strongly influences the health of many communities [1]. Thus, it is one of the guaranteed human economic and social rights by the majority of sovereign states in the world. Poor quality water is a major contributor to food- and waterborne diseases responsible for high mortality in developing countries, especially for children under 5 years old [2,3]. It has been associated with diseases such as diarrhea, cholera, dysentery, hepatitis A, typhoid and polio [4,5]. The impact of inadequate and poor water quality is mainly hard felt in the arid and semi-arid areas, which are on the rise due to deforestation, land degradation and the impact of climate change [6]. More often, these areas face torrential rains, massive flooding, sand mining and harvesting, leading to poor quality water, while other times prolonged droughts cause inadequate water supply [7–9].



It is estimated that by 2025, more than half of the world's population will experience water stress driven mainly by the impacts of climate change [10].

Sand dams are a frequently used technology for water harvesting and storage among the agro-pastoralists in arid and semi-arid lands (ASALs) [11,12]. It has also been suggested as one of the means of enhancing adaptation of these regions to climate change [13]. A sand dam is an impermeable physical barrier constructed across an ephemeral river to retain sand, which in turn acts as storage for rainwater. Due to the high rate of evapotranspiration found in the ASALs, open water sources such as dams and water pans are not reliable, making sand dams more suitable structures for water storage. Water can be obtained from the sand dams either through scooping holes or shallow wells [14]. Being key water resources during the dry seasons for the people and livestock in the ASALs of Kenya, it is critical to frequently monitor their water quality to guarantee access to quality water as envisioned in SDGs as well as informing water quality mitigation measures [15].

A number of studies have assessed various aspects of sand dams in Kenya, though this is limited mainly to the eastern region, with the majority looking at coliforms [16,17], pH, turbidity, heavy metals and electro-conductivity/salinity [17–20]. Additionally, there are few studies which have evaluated the hydrology of sand dams and their utilization [11,13,21]. Thus, despite most of the sand dams being prone to agro-based contamination, especially with nitrates ($NO_3$) and phosphates ($PO_4$) from applied fertilizers, and animal and human wastes, there is limited information on the distribution of these pollutants in existing sand dams [22,23]. The presence of some of these pollutants in drinking water, especially the two nutrients ($NO_3$ and $PO_4$), could also lead to a microbial population explosion, hence making portable water unsuitable for direct consumption, or increase the cost of water treatment [24].

Particles such as sand, silt, clay and organic matter retained within the sand dam influence both the quantity and quality of stored water [16,21,25]. Deposition and composition of these particles within the sand dams are expected to vary spatially and temporally because of their size, height of the spillway, water velocity and exposure time to degradation. Although retained particles are expected to influence quality of water over space and time, little is documented about it [16]. Therefore, this study aims at elucidating the chemical and microbial water quality as influenced by age of sand dams and position of scooping holes in degraded semi-arid environments. We hypothesized that (i) quality of water extracted from the sand dams (as measured by chemical and microbial properties of water) will improve with increasing distance from the sand dam wall, and (ii) quality of water extracted from the sand dams will improve with increasing age of the sand dams.

## 2. Materials and Methods

### 2.1. Study Site Description

The study was conducted in Chepareria, West Pokot County in Kenya. The area is located between latitude 1°15′ N to 1°55′ N and longitude 35°7′ E to 35°27′ E. The area is undulating, semi-arid with several hills and dominated by metamorphic rocks such as schist (Figure 1), which makes the area very susceptible to erosion. The mean annual rainfall is about 600 mm distributed across two rainy seasons, with the first season occurring between March and May and the second season between August and November. In 2021, the dry season began in September, and hence we conducted our study in mid-October 2021, which was the beginning of water scarcity in the area.

### 2.2. Selection of Sand Dams to Be Used in the Study

Sand dams were purposefully sampled based on water availability during the dry season. These were grouped into four categories: less than a year old, 1–9 years old, 10–20 years old and more than 20 years old. Under each category, three replications were selected to increase validity of the results. A total of 18 sand dams were sampled. Key informants' interviews were conducted to identify areas considered by the communities to have the observable differences in water quality. Two points were thus chosen. The first

point (hereafter referred to as A) was located 1 m from the sand dam wall, while the second one (point B) was located a further 20 m from point A. Scooping holes were dug into the sand, and existing water was scooped out using sterilized containers and discarded to let fresh water collect. This was to enable collection of clear water samples similar to what the farmers fetch for their domestic use [16]. Water samples were aseptically sampled using pre-sterilized containers. Before sampling, the containers were thoroughly rinsed with water from the scooping well.

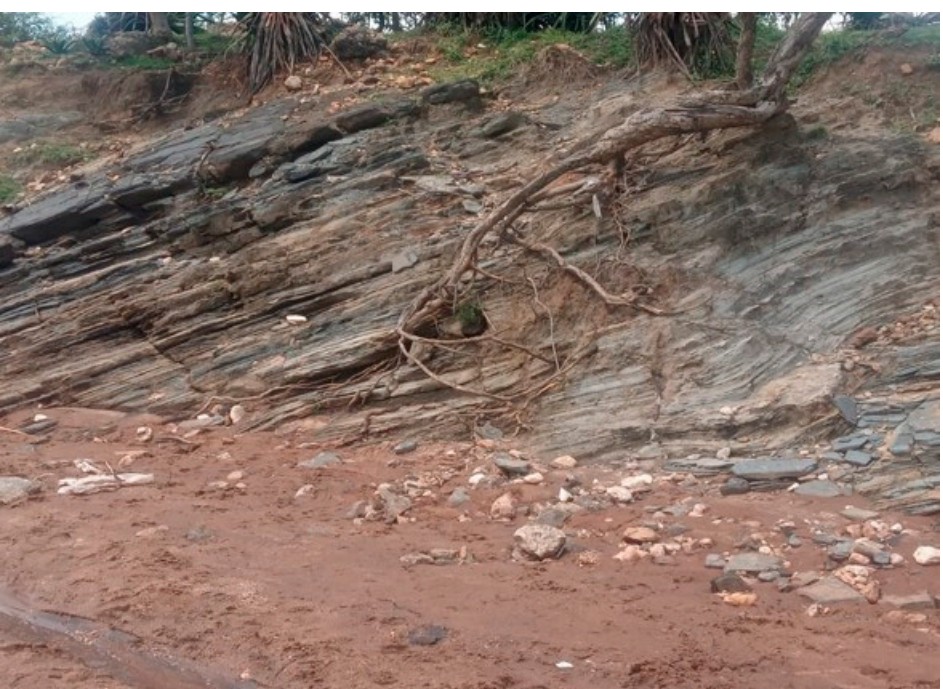

**Figure 1.** Schist rocks adjacent to a sand dam in Chepareria, West Pokot.

*2.3. Assessing Quality of Water Abstracted from the Scooping Holes*

Water quality parameters tested included salinity, total dissolved solutes (TDS), pH, thermotolerant and fecal coliforms, $NO_3^-$, $PO_4^{3-}$, $Fe^{3+}$ and water hardness. Salinity, TDS and pH were tested in situ using a portable YSI 556 multi-probe water quality meter multi-parameter water analyzer (YSI Environmental, Yellow Springs, OH, USA). The meter has a resolution of 0.1 mm Hg. The analyzer was calibrated to three points using buffer 7, followed by buffers 4 and 10, for pH readings [26]. Water samples from each sampling point were sampled using sterilized bottles and stored in a cooler ice box before being transported to the laboratory within 48 h for analysis. Two-stage analysis (presence–absence and multiple-tube methods) was used to confirm presence of fecal coliforms and quantification of thermotolerant coliforms according to the World Health Organization (WHO) guidelines [27]. In addition, $NO_3^-$, $PO_4^{3-}$, $Fe^{3+}$ and water hardness were analyzed according to American Public Health Association [28] laboratory manual.

*2.4. Data Analysis*

As part of data management, all data were subjected to the Hampel filter in R to check for outliers. The Hampel filter considers outliers as values outside the interval (*I*) formed by the *median* $\pm$ 3 *median* absolute deviations (*MAD*) such that

$$I = [median - 3 \times MAD; \; median + 3 \times MAD] \tag{1}$$

The data were also checked for normality using the Shapiro–Wilk test [29]. Data on TTC were log transformed before analysis of variance. Chemical and microbial properties data were subjected to analyses of variance (ANOVA) using R statistical software

(version 4.2.0) [30]. Thermotolerant coliform data were modeled using generalized linear mixed models as a function of sand dam age and location of scooping holes using the package lme4 in R [31]. Negative binomial regression was chosen as an extension of the Poisson distribution when analyzing coliform data. The best-fitting models were chosen based on the lowest Akaike Information Criterion (AIC). Where significant effects of sand dams and location of scooping holes were noted, means were separated using Tukey's Honest Significant Difference (HSD) test at $p < 0.05$.

## 3. Results

*Chemical and Microbial Properties of Water as Influenced by Sand Dam Age and Location of Scooping Holes*

Generally, the age of the sand dam and location of the scooping holes had varied effect on chemical properties of the water extracted from the sand dams (Figure 2). For instance, differences in pH were only observed between scooping holes in recently constructed sand dams (<1 year), where scooping holes near (well A) the sand dam wall had lower pH values (8.5) than holes scooped a distance (well B) from the sand dam wall (9.2). However, the age of sand dams had no significant effect on the pH of the water (Figure 2a). Similarly, the content of $NO_3^-$ in the water was only significant between the locations of the scooping holes within sand dams that were less than a year old. Here, water from scooping holes adjacent to the sand dam wall had significantly higher $NO_3^-$-N content (0.8 mg $L^{-1}$) compared to those located away from the dam wall (0.4 mg $L^{-1}$). The age of sand dams had no significant effect on the water $NO_3^-$ content (Figure 2b). In contrast, differences in Fe content in the water were significant in two age groups: the relatively recent sand dams (1–9 years) and the old ones (>20 years). In both age groups, the content of Fe in the water was lower in scooping holes near the sand dam wall than away, but the magnitude of differences differed greatly within and between these two age groups (Figure 2d). In 1–9-year-old sand dams, water extracted from holes near the sand dam wall had an average value of 1.6 mg $L^{-1}$ compared to holes away from the wall with values averaging 7.5 mg $L^{-1}$. This translates to about five times lower Fe content in water near the sand dam wall than away from it. In sand dams that were over 20 years old, however, water extracted from the holes near the dam walls had two times lower Fe content with an average of 0.5 mg $L^{-1}$, compared to holes located away from the dam wall with an average of 1.0 mg $L^{-1}$. The age of sand dams and the location of scooping holes had no significant effect on the content of $PO_4^{3-}$ (Figure 2c).

Contrary to chemical properties of the water, the age of sand dam had significantly greater effect on water total dissolved solutes, salinity, hardness and thermotolerant coliforms than the location of the scooping holes (Figure 3). Generally, both TDS and TTC decreased with the increasing age of the sand dams. For example, water from recently constructed sand dams had significantly high total dissolvable solutes (TDS) with an average value of 100.3 mg $L^{-1}$, which was 28.2 mg $L^{-1}$ more than what was obtained in water extracted from sand dams aged between 1 and 9 years, 25.9 mg $L^{-1}$ for sand dames aged between 10 and 20 years and 26.6 mg $L^{-1}$ in water extracted from sand dams that were more than 20 years old. However, there were no significant difference either between the three age groups (1–9 years, 10–20 years and >20 years) or between the location of scooping holes (Figure 3a). Similar differences were observed for water salinity and water hardness (Figure 3b,c, respectively). On the contrary, the age of the sand dam had little influence on thermotolerant coliforms; only the location of the scooping well had a significant effect. However, significant differences were observed only in sand dams aged between 1 and 9 years. Thermotolerant coliforms in water obtained from scooping holes adjacent to sand dam wall (well A) were greater than those reported in well B by more than 2500 CFU 100 mL$^{-1}$ of the water (Figure 3d).

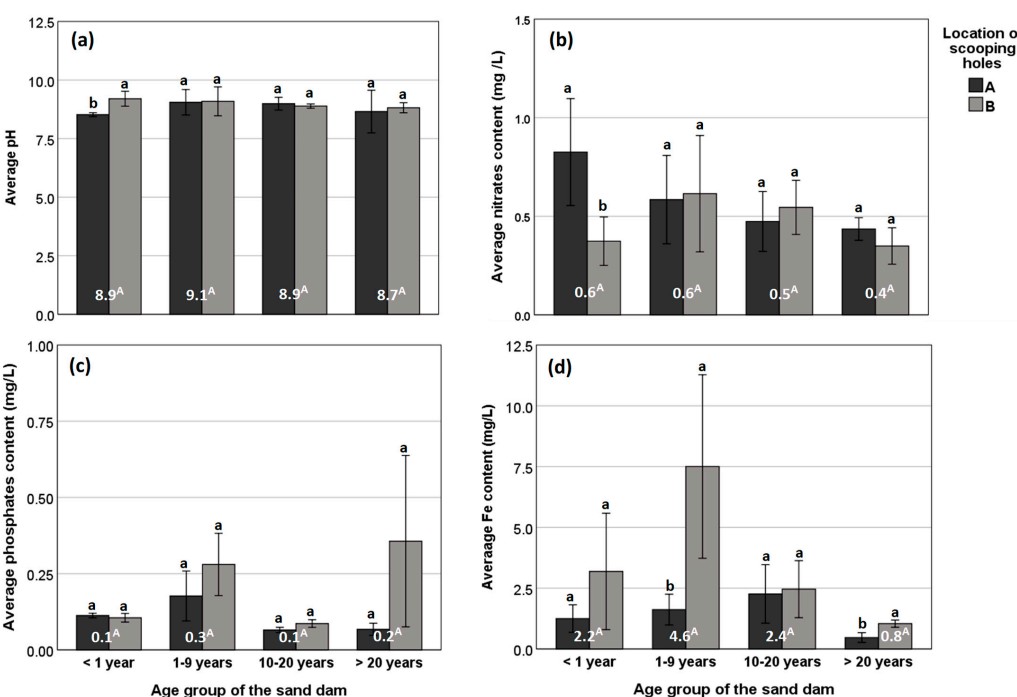

**Figure 2.** Chemical properties of water as affected by sand dam age and location of scooping holes; (**a**) pH, (**b**) nitrates, (**c**) phosphates and (**d**) iron. Values in text boxes separate means between age groups and those followed by different uppercase letters are significantly different at $p = 0.05$. Means with different lowercase letters indicate significant differences between the scooping holes. Bars above each category of the scooping holes represent the standard error of the mean.

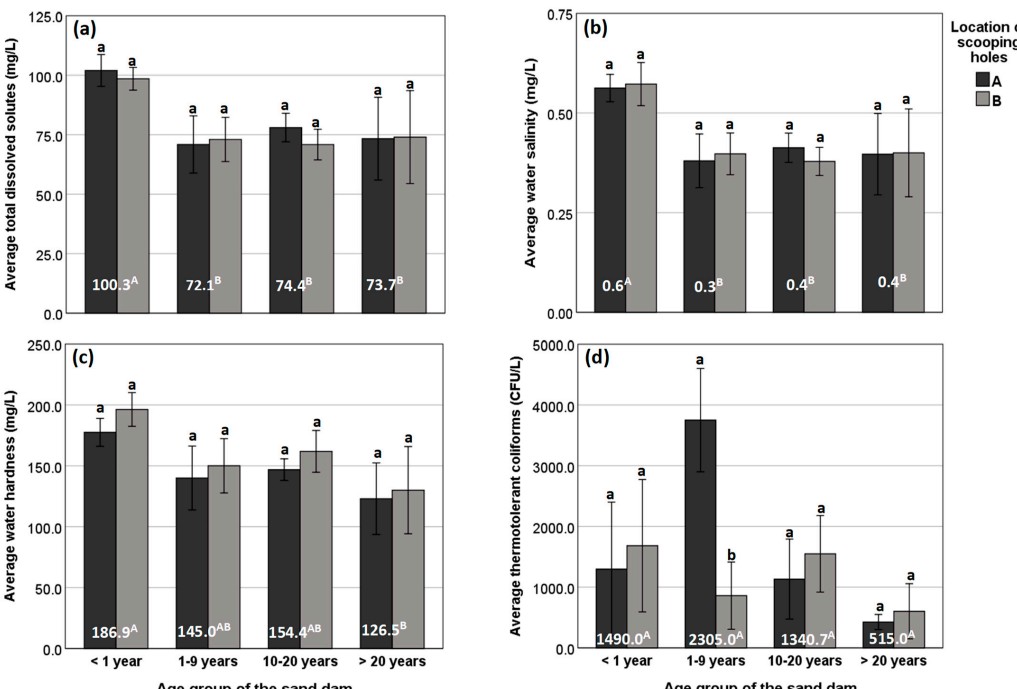

**Figure 3.** Chemical and microbial properties of water as affected by sand dam age and location of scooping holes; (**a**) total dissolved solutes, (**b**) salinity, (**c**) hardness and (**d**) thermotolerant coliforms. Values in text boxes separate means between age groups and those followed by different uppercase letters are significantly different at $p = 0.05$. Means with different lowercase letters indicate significant differences between the scooping holes. Bars above each category of the scooping holes represent the standard error of the mean.

## 4. Discussion

*Chemical and Microbial Properties of Water as Influenced by Sand Dam Age and Location of Scooping Holes*

Generally, the pH of water abstracted from scooping holes in all the age groups of the sand dams was high and exceeded the maximum recommended levels by the World Health Organization of 8.5 [32]. The pH values observed in our study were higher than what has been reported in other semi-arid environments in Kenya such as Kitui [19] and Makueni [20]. In both regions, Ndekezi et al. [19] and Ndunge et al. [20] reported the water obtained from scooping holes to have an average pH of 7.6, which was almost 1 pH unit lower than that observed in our study. In our study, the high pH can be attributed to the schist rocks that had been observed in the study area (Figure 1). Schist rocks contain appreciable amounts of carbonates and hydrogen carbonates of calcium, magnesium and sodium, which can contribute to the high pH [33,34]. Water with high pH and rich in calcium bicarbonate has been reported to reduce bone resorption, a critical process involved in the transfer of calcium from bone tissue to the blood [35]. Alkaline water also favors copper pitting corrosion through cathodic reaction [36] while reducing efficiency of water treatment through favoring the formation of scum [5,37]. The lack of significant influence of the age of sand dams on water pH reported in this study confirms the influence of the rock as the key determinant of pH.

Nitrates and phosphates contents were low in scooping holes across all the age groups of the sand dams compared to the recommended levels by the World Health Organization and Kenya Bureau of Standards of 10 and 2.2 mg $L^{-1}$, respectively [32,38]. The contents reported in this study were similar to those obtained by Ndunge et al. [20], where nitrates ranged between 0.2 and 2.6 mg $L^{-1}$, while phosphates were reported to be below 0.2 mg $L^{-1}$. The major source of these two nutrients in water is through sediments eroded from agricultural farms or from sewage effluents [39]. Nonetheless, the low contents could be attributed to possible pollution from fecal materials, mainly from livestock, and organic materials retained within the sand dam. In addition, formation of $CaPO_4$, which precipitates out of the water as insoluble solids could further reduce the content of phosphates. The nitrates also decreased with the age of the sand dam, which could be an indication that the stabilization of the dam could have been contributing to better quality water. This could be attributed to N uptake by the vegetation regeneration in the sand dams over time. Regeneration of vegetation around the sand dams could reduce the nitrate content. For instance, water extracted from scooping holes adjacent to sand dam walls had higher nitrate content than that from scooping holes away from the dam wall. The mode of sand dam natural stabilization usually starts from the far end and the sides toward the center of the sand dam. Thus, the vegetation growth around the sand dam could contribute to the observed spatial differences in nitrate content. Moreover, the higher amounts of sediments deposited adjacent to the sand dam wall during the initial stages of sand dam filling may lead to accumulation of more organic materials near the dam wall than away.

Generally, Fe content in water abstracted from scooping holes across all age groups of sand dams exceeded WHO recommendations of 0.3 mg $L^{-1}$ [32]. High content of Fe in drinking water could lead to conditions such as fibrosis [40,41]. However, pathogenesis of Fe-induced fibrosis is not well understood, nor has the critical Fe content in drinking water that could damage cells been established. In addition, high Fe content in drinking water reduces the aesthetic water quality by affecting its taste, smell and formation of microfilms in pipes and containers [5]. Similar amounts of Fe to those reported in this study were reported in Kitui [19] and in Makueni County [20]. In Kitui, Ndekezi et al. [19] reported an average Fe content of 1.9 mg $L^{-1}$ from water samples obtained from scooping holes. This was 0.6 mg $L^{-1}$ lower than what was reported in this study. Ndunge et al. [20], on the other hand, reported an average Fe content of 1.5 mg $L^{-1}$. In our study, Fe content decreased with sand dam age, an indication of a stabilization of the sand dams and the improvement of water quality with age.

Water from scooping holes across all age groups of sand dams had TDS within the acceptable range of <1000 mg/L based on WHO standards [32]. The values we obtained from our study were also lower than what has been reported in other similar studies. For example, in their study, Ndekezi et al. [19] reported an average TDS value of about 500 mg $L^{-1}$, which was 400 mg $L^{-1}$ higher than the highest average readings obtained in the current study. In another study within the same region, Kitheka [18] reported TDS values of up to 3320 mg $L^{-1}$ for the water scooped from sand dams in the river Thua in Kitui County. However, the author did not separate the TDS readings of the dry season from the wet season, despite alluding to an influence of seasons in that study. Thus, the high TDS in that study could have been as a result of taking readings during extremely dry periods since the dissolved inorganic and organic substances tend to increase during the dry season as more water is lost through evapotranspiration [42,43]. The TDS values also decreased with age of the sand dam, an indication of sand dam stabilization. Reduction in TDS in the water with stabilization of the sand dams could be of significance in protecting the community in the study area since TDS levels greater than 1000 mg $L^{-1}$ could contribute to cardiovascular diseases, diarrhea and abdominal pain [4].

Salinity levels were low in our study area, being less than 1% in all the sand dams. This was more than 10 times lower than what was reported by Kitheka [18] in Kitui County. Water hardness levels were within the allowable range of 100–300 mg $L^{-1}$ based on WHO standards [32]. Both salinity and hardness of the water reduced with the age of the sand dam.

Based on thermotolerant coliforms, the water abstracted from all the sand dams was not fit for human consumption without any form of treatment since no coliforms should be traced according to Kenya Bureau of Standards (KEBS) standards [38]. The presence of coliforms could be attributed to contamination with fecal matter from both animals and humans, since *Escherichia coli* was found in all the water samples. During collection of water samples, cattle and goats were observed drinking water from the old scooping holes that were made by the community members. The open nature of sand dams could exacerbate contamination of the water with fecal materials, as noted by Quinn et al. [16]. Several other studies have reported similarly high thermotolerant coliforms in water abstracted from scooping holes within semi-arid regions [16,17,20]. For example, Quinn et al. [16] observed TTC in a majority of water samples abstracted from scooping holes from Machakos and Makueni Counties, with approximately 20% of the samples containing more than 1000 CFU 100 $mL^{-1}$. In a more comprehensive study that covered three counties of Eastern Kenya (Machakos, Kitui and Makueni) across the rainy and dry seasons, Neufeld et al. [17] reported high TTC with an average of 800 CFU 100 $mL^{-1}$ in water samples obtained from scooping holes. The presence of high fecal coliforms poses a health risk in case of outbreak of diseases such as dysentery, diarrhea, hepatitis A, trachoma and skin infections [27]. Though sand dam age did not have significant effect on TTC, location of scooping holes had a significant effect on TTC for sand dams between age 1 and 9 years indicating a possible peak of decomposition of trapped organic materials and hence proliferation of coliforms. Stabilization of sand dams though encroaching vegetation could attract predators such as protozoa and bacteriophages, which have been shown to prey on Gram negative bacteria such as *E. coli*, as suggested by McCambridge and McMeekin [44] and Hobley et al. [45].

## 5. Conclusions

Sand dams are a critical source of water to people living in ASALs. It is thus important to monitor the quality of the water stored in the sand dams to ensure it meets the recommended minimum threshold for potable water. This study elucidated how chemical and biological water quality is influenced by the age of the sand dam and the location of the scooping holes. The age of the sand dam only influenced selected water chemical qualities (total dissolved solutes, water hardness and salinity). Position of the scooping holes significantly influenced a few chemical (pH, nitrates and $Fe^{3+}$) and microbial prop-

erties (thermotolerant coliforms). Some of the physico-chemical water parameters such as water hardness, $Fe^{3+}$ and pH exceeded the maximum allowable levels, which reduces water aesthetic quality. Additionally, water from the sampled sand dams was biologically unfit for direct consumption. To manage the high population of fecal coliforms, water from the sand dams should be treated through locally available methods such as boiling and chlorination before drinking. Protecting water abstraction points through the establishment of shallow wells with appropriate aprons has been shown to reduce microbial contamination through filtering underground water while providing a physical protection against surface contaminants. The limitation of this study was that the analysis of the samples was done in one laboratory. As a follow-up, we would recommend future studies to have more than one laboratory test for comparison.

**Author Contributions:** Conceptualization, H.C., S.K. (Solomon Kamau), W.N. and K.M.; methodology, H.C., S.K. (Solomon Kamau), W.N. and K.M.; software, H.C. and S.K. (Solomon Kamau); validation, H.C., S.K. (Solomon Kamau), W.N. and K.M.; formal analysis, H.C.; investigation, H.C., S.K. (Solomon Kamau), W.N. and K.M.; resources, W.N.; data curation, H.C.; writing—original draft preparation, H.C.; writing—review and editing, W.N., B.A., S.K. (Syphyline Kebeney), F.W. and J.M.; visualization, H.C., S.K. (Solomon Kamau) and J.M.; supervision, S.K. (Solomon Kamau), W.N. and K.M.; project administration, W.N., B.A., S.K. (Syphyline Kebeney) and F.W.; funding acquisition, W.N., B.A., S.K. (Syphyline Kebeney) and F.W. All authors have read and agreed to the published version of the manuscript.

**Funding:** The study was funded by the McKnight Foundation Grant No. 19-135.

**Data Availability Statement:** The data presented in this study are available on request from the corresponding author.

**Acknowledgments:** We are greatly indebted to the McKnight Foundation for funding this research through Drylands Farmer Research Network (FRN) Project. Special gratitude to Thomas Lokoriongor and Henry Lubanga for their invaluable assistance in collecting water samples and laboratory analysis.

**Conflicts of Interest:** The authors declare no conflict of interest.

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
