# Peer review of "Spatial–Temporal Influence of Sand Dams on Chemical and Microbial Properties of Water from Scooping Holes in Degraded Semi-Arid Regions"

_water, doi:10.3390/w15183207_

Round 1

Reviewer 1 Report

Manuscript title: Spatial-temporal influence of sand dams on chemical and mi- 2 crobial properties of water from scooping holes in degraded 3 semi-arid regions   

Manuscript id: water-2449680

Authors: Churu1 et al.

The manuscript is particularly strong regarding the less studied topic and the experimental setup on sand dams and the impact on water ……. The manuscript regarding the topic and results presented is of interest to the environmental science community and revisions based on the comments below are recommended before considering for publication.

Major comments

·         Insufficient Abstract: In the abstract, the main aim and background of the manuscript are missing, the current version it only highlights the result. In addition, it would be even better to have a sentence as a future perspective.

·         Line 52-58, the aim or hypothesis of the study is clear, however, the approach is missing ….

·         Lake of scientific literature to support the statements and findings throughout the manuscript…... I have made some suggestions for that and more need it….

·         More information is needed for ALL TABLE captions and define the abbreviation and units that are used. And adjust the significant figures for the table and manuscript.

·         I have a major concern about the results and discussion section. The authors describe the results and compare the results with previous studies, however, insight mechanisms are still insufficient.

·         This section is repeating information already presented and explaining things in an unnecessarily complicated way. The quality of the manuscript would benefit from the whole section being condensed.

Detailed comments:

Abstract

If the unit/abbreviation is not mentioned before, consider defining the abbreviation when mentioned for the first time.

Introduction:

Line 37-47, 59-66: These are rather long sentences, better to break them down into more sentences.

Line 58: A reference is needed here, for example, you can use: https://doi.org/10.1016/j.oneear.2021.05.010

Line 48-54: A complicated sentence, please revise and check the grammar

Line 70: A reference is needed here, for example, you can use: https://doi.org/10.1016/j.watres.2021.117610

Line 73: A reference is needed here

In MM section

Literature references are missing for all sub-section. It would be better to cite the references that the procedure adopted.

Additional info is needed for the table caption, most importantly significant figures.

In MM section, what is the quality control (QC) data? There is no mention of the QC.

In general, how many times you’ve recorded the data,? duplicate? Triplicate?..... what you mentioned in the text is not clear, please elaborate more on this

2.5. Statistical Analysis

How the comparison was made between the treatments? Ad see my comment for Figures

R&D section

These sections are repeating information already presented and explain things in an unnecessarily complicated way. The quality of the manuscript would benefit from the whole section being condensed, Line 131-162, Line 194-214, Line 250-272….

Figure 1. How comparing the treatments and assigning the letter for the statistical difference is confusing. For example, Figure 1 a: How you can have a, a, a, a, a ? Does this mean that the data is not significant? Please elaborate more, or consider changing the format

Conclusion

I believe there are other important conclusions that could be made from this study…. And the future perspectives for the following research are highly crucial here.

·         Grammar and punctuation issuers need to be addressed. I have selected/mentioned some as examples.

Author Response

Detailed comments:

Comment 1:

Abstract

If the unit/abbreviation is not mentioned before, consider defining the abbreviation when mentioned for the first time.

Response

Comment 2

Introduction:

Line 37-47, 59-66: These are rather long sentences, better to break them down into more sentences.

Response

Line 58: A reference is needed here, for example, you can use: https://doi.org/10.1016/j.oneear.2021.05.010

Line 48-54: A complicated sentence, please revise and check the grammar

Response

Suggestion to improve Line 48 and 54 have been effected.

Line 70: A reference is needed here, for example, you can use: https://doi.org/10.1016/j.watres.2021.117610

Line 73: A reference is needed here

In MM section

Literature references are missing for all sub-section. It would be better to cite the references that the procedure adopted.

Additional info is needed for the table caption, most importantly significant figures.

In MM section, what is the quality control (QC) data? There is no mention of the QC.

In general, how many times you’ve recorded the data,? duplicate? Triplicate?..... what you mentioned in the text is not clear, please elaborate more on this

 Response

We had three replicates. This has been addressed in the text.

2.5. Statistical Analysis

How the comparison was made between the treatments? Ad see my comment for Figures

R&D section

These sections are repeating information already presented and explain things in an unnecessarily complicated way. The quality of the manuscript would benefit from the whole section being condensed, Line 131-162, Line 194-214, Line 250-272….

Figure 1. How comparing the treatments and assigning the letter for the statistical difference is confusing. For example, Figure 1 a: How you can have a, a, a, a, a ? Does this mean that the data is not significant? Please elaborate more, or consider changing the format

Response

Addressed effectively as shown in the manuscript.

Conclusion

I believe there are other important conclusions that could be made from this study…. And the future perspectives for the following research are highly crucial here.

Response

We have added more contents in the conclusion. In respect to the future perspectives, we already have these conclusions “To manage high population of faecal coliforms, water from the sand dams should be treated through locally available methods such as boiling and chlorination before drinking. Protecting water abstraction points through establishment of shallow wells with appropriate aprons has been shown to reduce microbial contamination through filtering underground water while providing a physical protection against surface contaminants”. 

Comments on the Quality of English Language, Grammar and punctuation issuers need to be addressed. I have selected/mentioned some as examples.

Response

We have addressed a few as indicated in the manuscript. Regrettably, we were unable to follow those where references was given in form of lines.

Author Response

Reviewer 1

Comment 1; Change of chemical symbols for nitrates and phosphates NO-3, PO3-4

Response

This has been effected throughout the manuscript. We have also modified the same for Fe to read “Fe3+”.

Comment number 2

Lines 108 and 109: Model of the pH meter used, the test method used and resolution

Response:

This has been added.

Comment 3

Use of parametric and non-parametric tests

Response

The use of Shapiro test was to test for normal distribution of the data before conducting the ANOVA. Data that were abnormally distributed such as TTC were transformed before analysis.

Comment 4

The error bars descriptions for Fig 2 and 3:

Response

The error bars were for SE, standard error of the mean

Comment 5:

Possible flaw in the manuscript in respect to consideration on heterogeneity

Response:

Indeed these are possible sources of error in data collection. However, to cure this, we ensured we replication of sampled sand dams were done. A replicate of 3 sand dams per every category was done. We also checked for reproducibility in the means by looking at the coefficient of variance (CV) to ensure that there were no outliers in the data.

We have added some lines to take care of this in the Materials and Methods section (2.2).

Round 2

Reviewer 1 Report

The revised manuscript has improved compared to the original version. The authors tried to address my questions as much as possible. I recommend the manuscript to be published!

Overall, the English language and grammar quality are good and acceptable.

Author Response

This is noted and greatly appreciated. Thanks for your invaluable feedback. It greatly helped improve the quality of the paper.

Reviewer 2 Report

Reviewer 1

Comment 1; Change of chemical symbols for nitrates and phosphates NO-3, PO3-4

Response

This has been effected throughout the manuscript. We have also modified the same for Fe to read “Fe3+”. Ok!

Comment number 2

Lines 108 and 109: Model of the pH meter used, the test method used and resolution

Response:

This has been added. Unfortunately, I did not find this information in the manuscript. I suggest including it.

Comment 3

Use of parametric and non-parametric tests

Response

The use of Shapiro test was to test for normal distribution of the data before conducting the ANOVA. Data that were abnormally distributed such as TTC were transformed before analysis. It's ok, as for the normality of the data; but what about the outliers’ test?

Comment 4

The error bars descriptions for Fig 2 and 3:

Response

The error bars were for SE, standard error of the mean. Unfortunately, I did not find this information in the manuscript. I suggest including it.

Comment 5:

Possible flaw in the manuscript in respect to consideration on heterogeneity

Response:

Indeed these are possible sources of error in data collection. However, to cure this, we ensured we replication of sampled sand dams were done. A replicate of 3 sand dams per every category was done. We also checked for reproducibility in the means by looking at the coefficient of variance (CV) to ensure that there were no outliers in the data.

We have added some lines to take care of this in the Materials and Methods section (2.2).

The concept of reproducibility is related to different laboratories, and I suppose that it is not applied here. Maybe, the authors want to say “intermediate precision”. On the other hand, I do not understand how the authors evaluated the absence of outliers using the coefficient of variance (CV). I suggest detailing more this part in the manuscript, not only in the response for the reviewer.

Finally, I suppose that authors are not used to separating sample variability from analytical variability. Therefore, I recommend that authors access this guide: Ramsey MH, Ellison SLR (2019) Measurement uncertainty arising from sampling: A guide to methods and approaches. Second Edition, Eurachem/CITAC Guide, Eurachem. Furthermore, this article can be useful: de Jesus Leite, V., de Oliveira, E. C., & Aucélio, R. Q. (2021). Impact of the sampling process on the measurement uncertainty, a case study: Physicochemical parameters in diesel. Accreditation and Quality Assurance, 26(1) https://doi.org/10.1007/s00769-020-01452-6

Finally, I suppose that authors are not used to separating sample variability from analytical variability. Therefore, I recommend that authors access this guide: Ramsey MH, Ellison SLR (2019) Measurement uncertainty arising from sampling: A guide to methods and approaches. Second Edition, Eurachem/CITAC Guide, Eurachem. Furthermore, this article can be useful: de Jesus Leite, V., de Oliveira, E. C., & Aucélio, R. Q. (2021). Impact of the sampling process on the measurement uncertainty, a case study: Physicochemical parameters in diesel. Accreditation and Quality Assurance, 26(1) https://doi.org/10.1007/s00769-020-01452-6

Author Response

HC: We appreciate the continued effort by the reviewer to make the manuscript better. We confirm that all the questions/suggestions by the reviewer have been addressed and where modifications are required, we have implemented this, and appear here in blue. Specific adjustments in the manuscript have been also provided here in italics and square brackets for ease of the review process.

Reviewer 1

Comment 1; Change of chemical symbols for nitrates and phosphates NO-3, PO3-4

Response

This has been effected throughout the manuscript. We have also modified the same for Fe to read “Fe3+”. Ok!

Comment number 2

Lines 108 and 109: Model of the pH meter used, the test method used and resolution

Response:

This has been added. Unfortunately, I did not find this information in the manuscript. I suggest including it.

HC: The information has been added from L. 113 to L. 114 of the revised manuscript. The section of the statement reads: [……and pH were done in-situ using a portable YSI 556 multi-probe water quality meter multi-parameter water analyzer (YSI Environmental, 45387 OH, USA)]

Comment 3

Use of parametric and non-parametric tests

Response

The use of Shapiro test was to test for normal distribution of the data before conducting the ANOVA. Data that were abnormally distributed such as TTC were transformed before analysisIt's ok, as for the normality of the data; but what about the outliers’ test?

HC: More information on the test for outliers has been added from L. 124 to L. 127 of the revised manuscript. [As part of data management, all data was subjected to the Hampel filter in R to check for outliers. The Hampel filter considers outliers as values outside the interval (I) formed by the median ± 3 median absolute deviations (MAD) such that; I = [??????−3.?????????+3.???]]

Comment 4

The error bars descriptions for Fig 2 and 3:

Response

The error bars were for SE, standard error of the mean. Unfortunately, I did not find this information in the manuscript. I suggest including it.

 HC: The information has been added at the footnotes of Figure 2 L. 184 and Figure 3 L. 190 to 191 of the revised manuscript. The statement reads: [Bars above each category of the scooping holes represent the standard error of the mean]

Comment 5:

Possible flaw in the manuscript in respect to consideration on heterogeneity

The concept of reproducibility is related to different laboratories, and I suppose that it is not applied here. Maybe, the authors want to say “intermediate precision”. On the other hand, I do not understand how the authors evaluated the absence of outliers using the coefficient of variance (CV). I suggest detailing more this part in the manuscript, not only in the response for the reviewer.

Finally, I suppose that authors are not used to separating sample variability from analytical variability. Therefore, I recommend that authors access this guide: Ramsey MH, Ellison SLR (2019) Measurement uncertainty arising from sampling: A guide to methods and approaches. Second Edition, Eurachem/CITAC Guide, Eurachem. Furthermore, this article can be useful: de Jesus Leite, V., de Oliveira, E. C., & Aucélio, R. Q. (2021). Impact of the sampling process on the measurement uncertainty, a case study: Physicochemical parameters in diesel. Accreditation and Quality Assurance, 26(1) https://doi.org/10.1007/s00769-020-01452-6

HC: We request for understanding from the reviewer about the misrepresentation of facts in the response to the earlier review. This could have arose from the fact that the comment did not give much details on the direction of what the reviewer was referring to as ‘heterogeneity’. The correct way of checking for outliers is not coefficient of variation, but there are tests which are useful in this process. As stated above, we have added the correct information in the manuscript on the correct test we used for checking for outliers as can be seen from L. 124 to L. 127 of the revised manuscript, which reads [As part of data management, all data was subjected to the Hampel filter in R to check for outliers. The Hampel filter considers outliers as values outside the interval (I) formed by the median ± 3 median absolute deviations (MAD) such that; I = [??????−3.?????????+3.???]].

We also appreciate the reviewer for providing these insightful references. In our study unfortunately, we were not able to test the samples collected in several laboratories. This was partly due to the inadequate resources that would allow us to do the tests in different laboratories and partly due to the distance it would take for us to send the samples to the next nearest accredited laboratory, which is over 400 km from the study site. That is why we went for hedging against possible analytical variability by increasing sampling effort. Here, we not only replicated the samples during collection, but we also analyzed the samples in triplicates.

Round 3

Reviewer 2 Report

Comment 5: To address this issue, I suggest including this sentence in the manuscript: "This study considered only the analytical process, not taking into account the variability arising from sampling."